# Microbial Community and Fermentation Dynamics of Corn Silage Prepared with Heat-Resistant Lactic Acid Bacteria in a Hot Environment

**DOI:** 10.3390/microorganisms8050719

**Published:** 2020-05-12

**Authors:** Hao Guan, Yang Shuai, Yanhong Yan, Qifan Ran, Xia Wang, Dandan Li, Yimin Cai, Xinquan Zhang

**Affiliations:** 1College of Animal Science and Technology, Sichuan Agricultural University, Chengdu 6111130, China; 2Institute of Grass Science, Chongqing Academy of Animal Husbandry, Chongqing 402460, China; 3Japan International Research Center for Agricultural Sciences (JIRCAS), Tsukuba, Ibaraki 305-8686, Japan

**Keywords:** bacteria community, fermentation dynamics, high temperature, lactic acid bacteria, corn silage

## Abstract

To develop a silage fermentation technique to adapt to global climate changes, the microbiome and fermentation dynamics of corn silage inoculated with heat-resistant lactic acid bacteria (LAB) under high-temperature conditions were studied. Corn was ensiled in laboratory silo, with and without two selected strains, *Lactobacillus salivarius* LS358 and *L. rhamnosus* LR753, two type strains *L. salivarius* ATCC 11741^T^ and *L. rhamnosus* ATCC 7469^T^. The ensiling temperatures were designed at 30 °C and 45 °C, and the sampling took place after 0, 3, 7, 14, and 60 days of fermentation. The higher pH and dry matter losses were observed in the silages stored at 45 °C compared to those stored at 30 °C. Silages inoculated with strains LS358 and LR753 at 30 °C had a lower ratio of lactic acid/acetic acid. The dominant bacterial genera gradually changed from *Pediococcus* and *Lactobacillus* to *Lactobacillus* in silages during ensiling at 30 °C, while the bacterial community became more complex and fragmented after 7 d of ensiling at 45 °C. The high temperatures significantly led to a transformation of the LAB population from homo-fermentation to hetero-fermentation. This study is the first to describe microbial population dynamics response to high temperature during corn ensiling, and the results indicate that *L. rhamnosus* 753 shows potential ability to improve silage fermentation in tropics and subtropics.

## 1. Introduction

Silage is the world’s important feed resource for ruminants [1]. Whole-plant corn has become the most widely used crop in the world for silage because of its high biological yield, suitable starch content, high water-soluble carbohydrates (WSC), and lower buffer energy value [2]. These characteristics not only make corn have the desirable fermentation profile, but also high nutrition. Although silage is very old technology, and people have gradually improved the production process through continuous research, with further demand for high-quality silage, there are still many challenges. In addition to controllable management factors, uncontrollable climate-related factors are the main reasons for the instability of silage quality in this hot area [3].

Due to the effects of global warming, extremely high temperatures have become a key factor affecting the quality of silage by accelerating the fermentation process and aerobic deterioration in tropical and subtropical regions [4]. It is generally known that lactic acid bacteria (LAB) are the key to drive forage fermentation to produce lactic acid and lower pH to achieve long-term preservation. However, lactic acid bacteria have optimum temperatures for growth and reproduction, usually not higher than 45 °C [2]. Additionally, in the initial stages of fermentation, due to plant’s continuous respiration and aerobic microbial activity when air is still present in the plant gap, the core temperature of the silage usually rises above 40 °C in temperate areas, while temperatures are usually higher in tropical and subtropical regions, which means these areas may suffer longer-term high-temperature effects on silage during summer [4]. 

Generally, to ensure that silage enters the fermentation stage dominated by LAB as soon as possible, LAB inoculants are commonly used in silage production [5]. Typical homofermentative LAB strain *Lactobacillus plantarum* can inhibit the growth of microbe in silage by accelerating the decrease of pH [6], while *L. buchneri* as most used heterofermentative LAB strain can produce acetic acid to inhibit the growth of yeast and mold, thus improving aerobic silage stability [7]. However, due to these two commercial LAB inoculants were screened from temperate regions [2], which were mainly used for cold-season forages and corn to produce silage [2,8,9], their effects in tropical and subtropical regions may be limited. Chen et al. [10] reported that commercial LAB (*L. plantarum* and *Pediococcus acidilactici*) had no positive effect on Ryegrass silage at 45 °C. The effect of inoculating commercial LAB (*L. plantarum* MTD-1) was also limited in Napier grass silage, which stored at 50 °C [11]. Guan et al. [12] found that the commercial *L. plantarum* not only failed to improve the silage quality but also accelerated the aerobic deterioration in corn silage stored at 35–40 °C. These studies indicate that commercial LAB, which mostly screened from temperate regions, have limited adaptability to high-temperature environments.

Silage is a fermentation process dominated and driven by microorganisms [13]. Obviously, we cannot satisfy the exploration on the microbial community in the silage fermentation process through traditional microorganism counting and culture-dependent strain isolation technology. With the rapid development of molecular technology, next-generation sequencing (NGS) technology has begun to be widely used in the research of silage microbial communities, which provided deep insight into the silage fermentation process [3,14,15]. However, most silage researches were stored at normal temperatures ranging from 20 °C to 30 °C; therefore, we do not know what and how microbial communities change in silage under the high-temperature conditions.

To develop a silage fermentation technique in response to global warming, the current study aimed to apply NGS to determine the dynamic (0, 3, 7, 14, 60 d of ensiling) changes of microbial community and fermentation profile of corn silage prepared with two screened heat-resistant LAB (*L. salivarius* 358 and *L. rhamnosus* 753) in a hot environment (30 °C and 45 °C).

## 2. Materials and Methods

### 2.1. Forage and Ensiling

The whole plant corn (*Zea mays* L. cultivar Yayu No.8) used in this experiment was harvested at the kernel half milk line on 27 June 2018, at the Farm of Sichuan Agricultural University in Chongzhou, Sichuan Province, China (N30°33′23.98″ E103°38′42.61″). The self-propelled forage harvester (4QZ-18A, Muzhe Brands, Heibei, China) was set to chop at a theoretical length of 15 mm. The freshly chopped, whole-crop corn used in this study contained 283.8 g kg^−1^ dry matter (DM), and the contents of the crude protein (CP), neutral detergent fiber (NDF), acid detergent fiber (ADF), and WSC were 59.31, 451.59, 222.39, and 177.78 g kg^−1^ DM, respectively. Two strains *L. salivarius* 358 (LS358) and *L. rhamnosus* 753 (LR753) were isolated from forage crops and silages in hot and humid areas of southwest China (Sichuan Province, Chongqing city, Guizhou Province). Both strains can grow at high-temperature and low pH conditions and produce more lactic acid than other inoculant strains and isolates in silage environments. According to the screening results of physiological and biochemical properties and small-scale silage fermentation tests of more than 200 LAB isolates, the LAB strains were selected for improving silage fermentation and inhibiting aerobic deterioration of silage [12]. The treatments consisted of sterile distilled water (Control), two selected strains *L. rhamnosus* 753 (LR753) and *L. salivarius* 358 (LS358), two type strains *L. salivarius* ATCC 11741 (LS) and *L. rhamnosus* ATCC 7469 (LR). Four LAB strains were inoculated as a single culture by using de Man, Rogosa and Sharpe (MRS) broth (CM 188, Land Bridge Brands, Beijing, China) and distilled to an equivalent of around 10^6^ colony-forming units cfu g^−1^ of fresh matter (FM), then sprayed evenly on the corn material and thoroughly mixed. One kg of the mixed material was filled in a sterile plastic bag (dimensions 300 mm × 400 mm, 2-ply 3 mil polyethylene, Aodeju Brands, Sichuan, China) and evacuated and sealed by an internal pumping vacuum packaging machine (DZ-600/2SD, Xinbo Brands, Zhejiang, China). Each treatment had three biological replicates, a total of 120 vacuum-bags were separated and stored in a dark environment at 30 °C and 45 °C, respectively. The fresh corn was taken for analysis of chemical composition and microbial community. Samples of 3, 7, 14, 60 d for ensiling were used for the analysis of the dynamic changes of microbial community and fermentation, and a sample from the 60 d was also used for the analysis of chemical composition. After opening all samples, the number of microorganisms was detected immediately, and two subsamples from each sample were taken quickly into −80 °C lab freezers (Thermo Fisher Scientific, Waltham, MA, USA) for DNA extraction and detection of relevant fermentation parameters.

### 2.2. Chemical, Fermentation and Microbial Counts Analysis

Three fresh samples and all 60-day silage samples were placed in an air-circulated oven at 65 °C for 72 h to test the DM. The weight and DM of the samples before and after 60 d of silage were used to calculate DM recovery (DMR). The dried samples were ground by grinder (CT293 Cyclotec^™^, FOSS Analytical A/S, Hilleroed, Denmark) and passed through a 1 mm mesh sieve for future chemical analysis. An Ankom 200 system (Ankom Technology Corporation, Fairport, NY, USA) was used to test NDF and ADF, and the test method was in accordance with the instructions. The CP was determined by the Kjeldahl method and WSC was measured by the thracenone-sulfuric acid method [16].

A 20 g of wet silage samples from 3, 7, 14, and 60 d were added to 180 mL of distilled water and stayed overnight in a refrigerator at 4 °C, and then filtered through two layers of cheesecloth to obtain the silage extract. The pH of the extract was tested using a pH meter (PHSJ-5; LEICI, Shanghai, China), and the remaining samples were used to test organic acids concentration and NH_3_-N. Organic acids (lactic acid, acetic acid, propionic acid, butyric acid) were analyzed by high-performance liquid chromatography (HPLC) with a UV detector (210 nm) and a column (KC-811, Shimadzu Co. Ltd., Kyoto, Japan), according to the method by Guan, et al. [3]. NH_3_-N was determined according to the method by Weatherburn [17].

The amount of LAB was quantified by De Man, Rogosa, Sharpe agar (CM 188, Land Bridge, Beijing, China), molds and yeasts counts were cultured on Potato Dextrose Agar (CM 123, Land Bridge, Beijing, China), and coliform bacteria were quantified using Violet Red Bile Agar (CM 115, Land Bridge, Beijing, China).

### 2.3. Bacterial Community Analysis

#### 2.3.1. DNA Extraction

The method of DNA extraction referred to Guan et al. [3] and Yan et al. [15]. A total of 50 g of frozen sample passed through a 4 mm sieve after freeze-drying and smashing. A subsample (5 g) was ball milled for 1 min at room temperature, and the total DNA was extracted via the TIANamp Bacteria DNA isolation kit (DP302-02, Tiangen, Beijing, China). All samples were purified via purification and recovery of the DNA kit column (DP214-02, Tiangen, Beijing, China), and then eluted in nuclease-free water. NanoDrop2000 was used to detect the purity and concentration of DNA. The qualified DNA samples were stored at −20 °C for future analysis.

#### 2.3.2. Sequencing

16S rRNA genes of distinct regions (16S V4) were amplified used the specific primers 515F (5′-GTTTCGGT GCCAGCMGCCGCGGTAA-3′) and 806R (5′-GCCAA TGGACTACHVGGGTWTCTAAT-3′) [3]. Samples with a bright main strip between 400–450 bp were chosen for further experiments. Sequencing libraries were generated and sequenced as previously described in Guan et al. [3].

#### 2.3.3. Sequences Analyses

Next-generation sequencing reads were assembled using FLASH (V1.2.7) [18]. According to the QIIME quality control process (V1.7.0) [19], low-quality reads were excluded. Chimeric sequences were removed by using the UCHIME algorithm to obtain final effective tags [20]. The Uparse software was used for sequence analyses (Uparse v7.0.1001) [20]. Operational taxonomic units (OTUs) were defined by a 97% similarity cutoff. To annotate taxonomic information, the representative sequences of each OTU were aligned to the Greengene Database [21] based on the RDP classifier algorithm (Version 2.2) [22]. Alpha diversity metrics (Observed-species, Chao1, Shannon, Simpson, ACE, and Good-coverage) and beta diversity metrics (weighted UniFrac and unweighted UniFrac) were calculated with the QIIME software (Version 1.7.0). PCoA analysis was conducted with R software (Version 2.15.3). The sequence data reported in this study have been submitted to the NCBI database (PRJNA606702).

### 2.4. Statistical Analyses

Microbe populations were estimated as colony-forming units (cfu) g^−1^ and were log-transformed prior to statistics. The statistical analyses were performed using the GLM procedure of SPSS v19.0. Tukey’s honest significant difference (HSD) test was employed, and significance was declared at *p* < 0.05.

## 3. Results

### 3.1. Fermentation Characteristics and Microbial Counts of Silage at Different Period

Fermentation characteristics and microbial counts of corn silage at different periods are shown in Table 1 and Figure 1. After 3 d of fermentation, pH values of all silages decreased sharply compared to forage and were all below 4.2. Silages stored at 45 °C had a slightly higher pH value than that of silages stored at 30 °C (*p* < 0.05), LS-treated silage had the highest pH at 45 °C, while LR-treated silage had the lowest pH at 30 °C (*p* < 0.05). Control silages at both temperatures had the highest NH_3_-N (g kg^−1^ TN) rates compared to the LAB inoculated silages (*p* < 0.05). Counts of LAB in all silages increased at 30 °C with large amounts of lactic acid production after three days of fermentation (*p* < 0.05). However, although high contents of lactic acid were found in all silages at 45 °C, lower LAB counts were observed compared to forage. The number of yeasts, molds, and coliforms decreased in all silages at both temperatures compared to forage, and there was no acetic acid produced after 3 d of fermentation. Temperature, inoculants, and their interaction had an extremely significant effect on pH, number of LAB, and yeast (*p* < 0.01).

After 7 d of fermentation (Table 1 and Figure 1), all silages at 45 °C had a higher pH than those at 30 °C (*p* < 0.05), the highest pH was found in the control silage at 45 °C (*p* < 0.05). NH_3_-N contents in all silages increased compared to silages ensiled for 3 d and ranged from 20.24 to 27.40 g kg^−1^ TN. Although the pH value of all silages did not change much, lactic acid contents increased in all silages compared to silages ensiled for 3 d. Acetic acid in all silages at 30 °C were observed after ensiling for 7 d, while only screened LAB (LS358 and LR753) inoculated silages observed acetic acid (AA), and the highest AA contents were found in LR753-inoculated silage stored at 30 °C (*p* < 0.05). LAB counts of controls at 30 °C increased while LAB-treated silages showed small decline compared to silages ensiled for 3 d. LAB counts of all silages at 45 °C still showed a downward trend. Mold was detected only in control silage at 45 °C, which was 4.54 log10 cfu g^−1^ of FM. Temperature, inoculants, and their interaction had an extremely significant effect on pH, AA contents, LA/AA, number of LAB, and yeast (*p* < 0.01).

As the fermentation time prolonged to 14 d, as shown in Table 1 and Figure 1, the pH of all silages had a slight decrease compared to the samples that were fermented for 7 d, except for the control silage at 45 °C, which also had the highest NH_3_-N contents with the control at 30 °C (*p* < 0.05). NH_3_-N (g kg^−1^ TN) of all groups increased with the extension of the fermentation process. The content of LA did not change much compared to the silages ensiled for 7 d, except for the LR-treated silage at 30 °C that increased significantly, and was found to be the highest (*p* < 0.05). The content of AA continued to increase, and the highest AA content appeared in LR753 treatments at 30 °C (*p* < 0.05), which reached 61.22 g kg^−1^ of DM. The number of LAB in all 30 °C treatments remained between 2.71 and 7.09 log10 cfu g^−1^ of FM, the highest number of LAB were observed in LR753 treatments at both temperatures. The number of yeasts continued to decrease, and there were still no mold and coliforms detected in all silages. Temperature, inoculants, and their interaction had an extremely significant effect (*p* < 0.01) on pH, AA contents, number of LAB, and yeast. 

At the 60 d of ending, the pH of all samples at both temperatures ranged from 3.81 to 4.06 (Table 1 and Figure 1), the highest pH was found in the control group at 45 °C (*p* < 0.05). The NH_3_-N (g kg^−1^ TN) continued to increase, but all silages did not exceed 50 g kg^−1^TN, the lowest NH_3_-N (g kg^−1^ TN) was observed in LR753 groups at 45 °C (*p* < 0.05). The contents of NH_3_-N (g kg^−1^ TN) and LA in all silages at 45 °C were higher than the same-treated silages at 30 °C, while silages at 30 °C had higher AA contents (*p* < 0.05). The number of LAB decreased in all groups at 30 °C compare to silages ensiled for 14 d, ranging from 4.15 to 4.66 log10 cfu g^−1^ of FM. There were no LAB, yeasts, mold, and coliforms detected in all silages at 45 °C. Temperature, inoculants, and their interaction had an extremely significant effect on pH, NH_3_-N (g kg^−1^ TN), and LA/AA (*p* < 0.01).

### 3.2. Chemical Composition of Corn Silages Ensiled for 60 d

Chemical composition of corn silage ensiled for 60 d at 30 °C and 45 °C are shown in Table 2. DM contents of all silages at 30 °C decreased compared to forage, while it increased in silages stored at 45 °C. However, DM recovery of silages at 30 °C were remarkably (*p* < 0.05) higher than that at 45 °C. CP contents of all silages increased compared to forage except for LS treatment at 45 °C. Contents of WSC decreased after 60 d of fermentation; however, silages at 45 °C had significantly (*p* < 0.05) higher WSC contents than silages at 30 °C. The NDF and ADF showed a small difference in all treatments at both temperatures. Temperature, inoculants, and their interaction had an extremely significant (*p* < 0.01) effect on DM, CP, and WSC.

### 3.3. Dynamic Changes of Microbial Communities

The taxonomic profile of the bacterial core microbiome varied among fresh forage and dynamic changes of corn silages at 30 °C and 45 °C are shown in Figure 2A,B. *Leuconostoc*, *Klebsiella* and *Lactococcus* spp. were the dominate LAB in fresh corn forage. The abundance of *Lactobacillus* and *Pediococcus* spp. increased significantly after 3 d of fermentation. Pediococci in silages at 45 °C was generally higher than that at 30 °C. Especially, LS358 and LR753treatments had a higher abundance of pediococci at both temperatures. The communities of bacteria in all silages at 30 °C ensiled for 7 d had a similar situation with that ensiled for 3 d, except for the abundance of lactobacilli, which increased a lot while pediococci decreased in LR753 treatments at 30 °C. However, the bacterial community dramatically changed compared to the 3 d of fermentation at 45 °C. The abundance of pediococci sharply decreased while a large number of bacteria with small abundance began to appear in silages ensiled for 7 d at 45 °C. Besides, *Bacillus* spp. presented in the control group and samples inoculated with LS358, and *Acetobacter* spp. was found in LS and LR-treated groups at 45 °C. As the fermentation time prolonged to 14 d, pediococci was still the dominate bacteria in the LS358-treated group, while the LR753 group had the highest abundance of *lactobacilli* at 30 °C. Novelty, a high abundance of acetobacter was found in all silages except for the LR753 treatment at 45 °C. At the ending of fermentation (60 d), *lactobacilli* were the prevalent bacteria in all silages at 30 °C, followed by acetobacter, while the highest abundance of lactobacilli was observed in LR753 and LR treatments. The bacterial communities of each group were more complex at 45 °C, the abundance of lactobacilli and acetobacter decreased compared to silages ensiled for 14 d, while the abundance of *Pectobaterium*, *Brevundimonas*, and *Bosea* spp. increased a lot. Besides, the *Bacillus* spp. became the dominant bacteria in LR753 treatment at 45 °C.

As shown in Table 3, the effect of inoculants and temperature on the microbiota community during ensiling and differences in Bray-Curtis distance were analyzed by permutational multivariate analysis of variance (Anosim and PERMANOVA). The temperature has been significantly (*p* < 0.05) affecting the bacterial community in corn silage from 3 d to 60 d, of which the greatest effect was observed at 60 d (R-value = 0.999). The significant (*p* < 0.05) effect (*p* < 0.05) of the inoculants on the bacterial community only lasted to d 7, and there was no significant effect (*p* > 0.05) at 14 and 60 d of ensiling. Moreover, the R-value < 0 of the inoculants was observed at 7, 14, and 60 d of ensiling.

In order to investigate the effects of temperature on the dynamic change of microbial community in corn silages, we pooled the samples under the same temperature and fermentation time into one group (Figure 3). There were 119 shared OTUs of all the samples at 30 °C, while 199 shared OTUs were observed at 45 °C. With the exception of forage and silages fermented for 3 d and fresh martial, the number of unique OTUs of silages ensiled for 7 d, 14 d, and 60 d at 45 °C was much higher than that at 30 °C. At both temperatures, the highest number of unique OTUs appeared in silages ensiled for 7 d, which were 208 unique OTUs in silages at 30 °C and 1036 unique OTUs at 45 °C, respectively. Besides, only four unique OTUs were found in forage compared with silages at 45 °C. The bacterial alpha diversities of the samples were evaluated via the Chao1 and Shannon indexes (Figure 4A,B). At 30 °C, bacterial diversity decreased with the fermentation process. However, the bacterial diversity of silages increased at 45 °C compared to forage, especially after 3 d of fermentation. The bacterial diversity of silages at the same fermentation time was found much higher at 45 °C than that at 30 °C.

## 4. Discussion

Wilkinson and Muck [23] reported that global warming had been firstly proposed as an important factor for future silage. Facing the possible, increasing temperature, not only the species of crops that could be used for silage may be reduced, but also the microbial community of silage would be affected. However, most laboratory silage studies were conducted between 20 and 30 °C, which was difficult to represent the actual situation of silage in hot areas. Therefore, research should be aimed at studying patterns of fermentation and aerobic spoilage at temperatures above 40 °C, with the expectation that these conditions will become more common, especially in tropical regions.

### 4.1. Dynamic Changes of Silage Fermentation Characteristics and Microbial Composition

McDonald et al. [2] indicated that the most optimum growth temperature for LAB found in silage was around 30 °C and generally did not grow at or over 45 °C. In this study, the pH and NH_3_-N (g kg^−1^TN) in the initial stage of the ensiling of all silage at 45 °C were higher than those at 30 °C, while the number of LAB was lower. After 60 d of fermentation, no LAB was observed at the detected level in all silages; this indicated that most LAB could adapt to 45 °C at the beginning of ensiling, but they could not survive after two weeks under such high temperature. Clostridia generally have a higher optimum growth temperature, typically 37 °C, and will grow at 45 °C [2]; therefore, high temperatures often allow to occur butyric acid fermentation and more proteolysis during ensiling, result in poor fermentation quality [10,24]. In the current study, no butyric acid was detected in silages during the whole period; however, NH_3_-N (g kg^−1^ TN) of all treatments at both temperatures increased, especially at 45 °C. The optimal growth temperature of Clostridium is usually higher than that of LAB, and it can grow normally at 45 °C [23], while the high WSC corn used in this experiment may promote lactic acid fermentation and inhibit the growth of clostridia. Plant and microbial proteolytic processes lead to changes in nitrogenous compounds in silages [25], the temperature may affect the activity of proteases. According to Kung Jr et al. [25], the ratio of lactic acid to acetic acid is commonly used as a qualitative indicator of fermentation; good silage fermentations usually have a ratio of these acids of about 2.5 to 3.0. However, after 60 d of fermentation, lower ratios were found in all silages at 30 °C, and two LR-treated silages at 45 °C in this study. This indicated that the temperature at 30 °C leads to the change of fermentation type from homofermentation to heterofermentation, especially after inoculated heterofermentative LAB, but the high temperature at 45 °C restricted the growth of the heterofermentative LAB, except for LR753-treated silages. The low ratio does not necessarily mean low quality silage. Since strain *L. buchneri* was reported to improve aerobic stability of silage [26], many experiments [27,28,29] examined it as a typical heterofermentative LAB in silage for increasing aerobic stability by producing acetic acid as a means to inhibit yeast growth; however, the effects are strain-specific [30,31] and dose-dependent [32,33]. Due to commercial *L. buchneri* was screened from temperate regions [2], which was mainly used for cold-season forages and corn in the United States and Europe [2,8,9], their effects in tropical and subtropical regions may be limited. Our previous study showed that strain LR753 isolated and screened from hot, and highly humid areas can significantly increase the aerobic stability of corn silage in a high-temperature environment [12]. In the present study, high acetic acid content in strain LR753-treated silage also showed the potential ability to improve the aerobic stability of corn silage at high temperatures.

Generally, the information obtained through the detection of these conventional fermentation data was very limited, so a deeper understanding of the dynamics of microbial communities during ensiling is highly required.

### 4.2. Dynamic Changes of Microbial Communities

Generally, epiphytic bacterial communities are highly related to forage types [34] and climate [4]. In our previous study, Guan et al. [3]investigated the microbiome of 96 corn fresh and silage samples in high temperature and humidity regions and reported that the genus *Weissella*, *Pseudomonas*, *Lactobacillus*, and *Leuconostoc* spp. were the most dominate epiphytic bacteria genera, while *Leuconostoc*, *Klebsiella*, and *Lactococcus* spp. were the most prevalent bacteria found in the current study. These results indicated that the epiphytic bacterial community was not only affected by forage and climate type, but it was more related to individual differences, and although *Lactobacillus* and *Lactococcus* spp. existed in fresh corn, they were usually not dominant bacteria. In this study, *Lactobacillus* and *Pediococcus* spp. began to dominate at 3 d of the ensiling, this was different from the report by Drouin et al. [35], which *Lactobacillus* spp. began to dominate at 14 d of fermentation in corn silage inoculated with *L. buchneri* and *L. hilgardii*. This may be because the higher temperature in the current study promoted the rapid propagation of LAB, which is consistent with the results reported by Keshri et al. [6] that *Lactobacillus* spp. began to dominate at 2 d of corn silage in the warm climate. *Lactobacillus* and *Pediococcus* spp. are common predominant bacteria genera in silage [13]. Especially, pediococci was often observed in the early stage of the ensiling, which was consistent with the result of pediococci that was the dominant bacteria from 3 to 7 d and almost disappeared at the terminal fermentation. However, the microbial community began to show significant differences at the two temperatures on the 7 d of fermentation. The dominance of pediococci began to decrease while the small abundance of bacteria began to appear in large numbers at 45 °C, which lasted until 60 d of ensiling.

Higher ensiling temperatures typically lead to a shift from homofermentative to heterfermentative LAB population [4]. Current studies not only confirmed this view but also indicated that such changes occurred at 7 d of ensiling. Generally, the high abundance of *Acetobacter* species was not common in the reported laboratory small-scale silage, while it was regularly detected in bunker-made corn silage [3]. However, acetobacter began to appear on the 7 d and dominated on the 14 d of fermentation at 45 °C, except for LR753-treated silage. These results indicated that acetobacter could also be found in well-sealed small-scale silage, and their presence is associated with high temperatures. Besides, LR753-treated silages at 45 °C had the highest abundance of lactobacilli, which partly because screened LR753 can well adapt to high-temperature conditions [12]. In recent years, *L. rhamnosus* has been reported as a silage additive, especially as heat-resistant inoculants [11,36]. However, this advantage did not extend to 60 d of ensiling at 45 °C, while bacterial composition has become more complex, traditional dominant LAB almost all disappeared, and some heat-resistant bacteria begin to grow. Instead, bacterial composition at 30 °C was more in line with silage conventions. The abundance of lactobacilli in all groups exceed 75%, LR753- and LR-treated silages had the highest lactobacilli and lowest acetobacter, which indicated that *L. rhamnosus* was more effective than other inoculants at 30 °C. Results from Anosim and PERMANOVA analysis showed the same finding, the temperature was the main factor affecting the bacterial community during ensiling, and achieved the greatest effect on the terminal of fermentation. While the R-value < 0 of the inoculants was observed at 14 and 60 d of ensiling, indicating that inoculants affected the bacterial community only lasted in the early two weeks.

### 4.3. Chemical Composition of Corn Silages Ensiled for 60 d

Higher ensiling temperatures typically lead to a shift from a homofermentative to a heterofermentative LAB community [4], which often leads to higher DM losses during ensiling [2]. Rees [37] reported DM losses of 1.7% for every 10 °C increase in temperature in laboratory-scale silo. In the present study, DMR in all treatment groups at 30 °C was much higher than at 45 °C, which agree with the result of Kim and Adesogan [38]. Silage is a microbial-based fermentation process, WSC as the substrate of microorganisms is an indicator of fermentation progress [2]. Due to those microorganisms in silage could hardly be detected at 45 °C, in the present study during the middle and later stages of fermentation, the WSC concentration in all treatments was quite high at 45 °C. The higher residual WSC concentrations of Hot vs. Cool silages agrees with the results of Colombatto*,* et al. [39] and Kim and Adesogan [38]. Wilson et al. [40] have reported that high temperatures increase the deposition of NDF, ADF, and lignin. However, no significant difference was observed in our study. This is partly because their experimental materials are originated from warm and cool areas, respectively. Higher temperatures increase the growth rate of plants and the activity of lignin synthase [41], thus increasing the differentiation of DM into more lignified tissues, while fresh materials in the present study were found similar.

## 5. Conclusions

The bacterial community of fresh whole-crop corn was found to be dominated by *Leuconostoc*, *Klebsiella*, and *Lactococcus* spp. Lactobacilli were the prevalent bacteria in all silages at 30 °C, followed by acetobacter, while the bacterial communities became more complex at 45 °C at 60 d of ensiling. High temperature affected the dynamic changes of microorganisms in corn silage and resulted in a shift from homofermentative to heterofermentative LAB community, which led to poor fermentation with more DM loss and higher NH_3_-N content. The heat-resistant strain *L. rhamnosus* 753 showed the potential possibility to respond to global warming for improving silage fermentation in tropics and subtropics. The bacterial monitoring dynamics by NGS technology is expected to better understand the relationships between microbial community and silage fermentation.

## Figures and Tables

**Figure 1 microorganisms-08-00719-f001:**
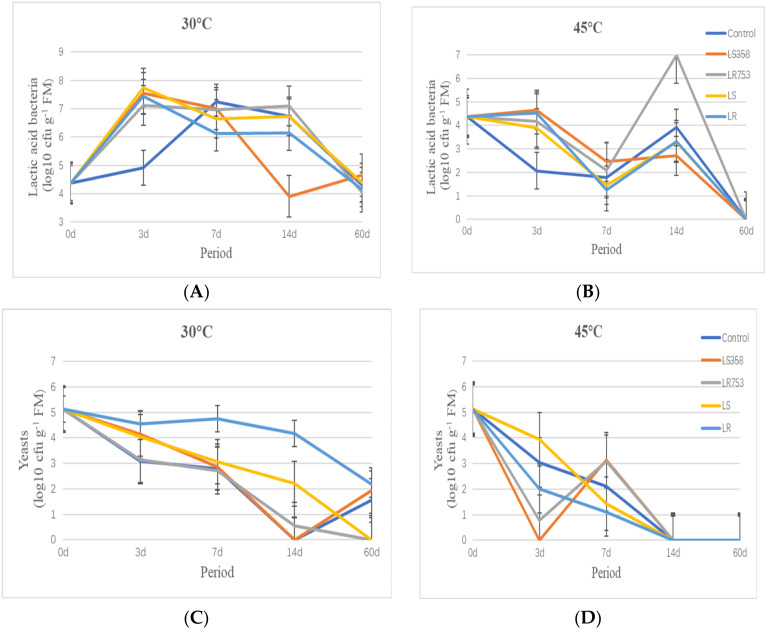
Dynamic microbial counts of corn silage prepared with two temperatures at different ensiling period. (**A**) Dynamic counts of lactic acid bacteria at 30 °C; (**B**) Dynamic counts of lactic acid bacteria at 45 °C; (**C**) Dynamic counts of yeast at 30 °C; (**D**) Dynamic counts of yeast at 45 °C. Note: LS358 =*Lactobacillus salivarius* 358; LR753 = *Lactobacillus rhamnosus* 753; LS = *Lactobacillus salivarius* ATCC 11741; LR = *Lactobacillus rhamnosus* ATCC 7469.

**Figure 2 microorganisms-08-00719-f002:**
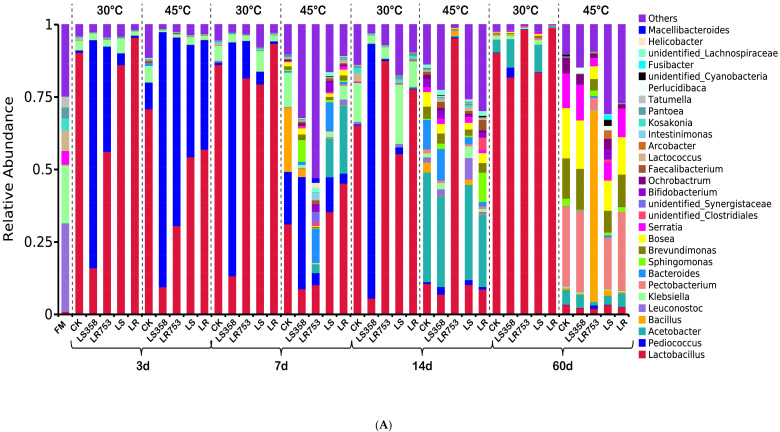
Taxonomic profile of the bacterial core microbiome varied among fresh forage and dynamic changes of corn silages at two temperatures at the genus level. (**A**) The relative abundance of the bacterial microbiome in every sample before and after ensiling; (**B**) Average values of silage samples for every period (FM: fresh material; 3 d; 7 d; 14 d; 60 d). Note: CK = Control; LS358 = *Lactobacillus salivarius* 358; LR753 = *Lactobacillus rhamnosus* 753; LS = *Lactobacillus salivarius* ATCC 11741; LR = *Lactobacillus rhamnosus* ATCC 7469.

**Figure 3 microorganisms-08-00719-f003:**
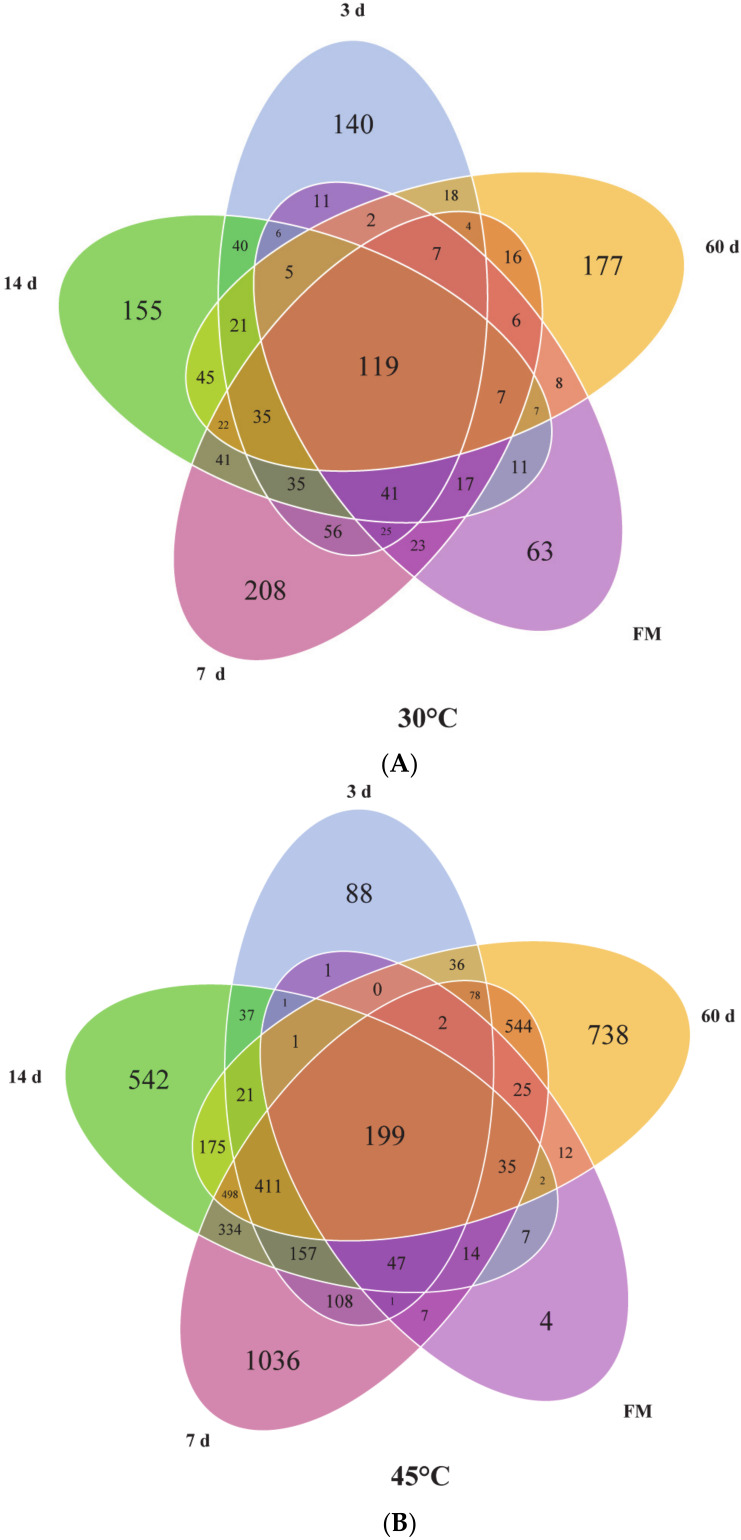
Venn diagram depicting unique or shared bacterial OTUs in silages for every period (FM: fresh material; 3 d; 7 d; 14 d; 60 d). (**A**) Venn diagram depicting unique or shared bacterial OTUs in silages for every period at 30 °C; (**B**) Venn diagram depicting unique or shared bacterial OTUs in silages for every period at 45 °C.

**Figure 4 microorganisms-08-00719-f004:**
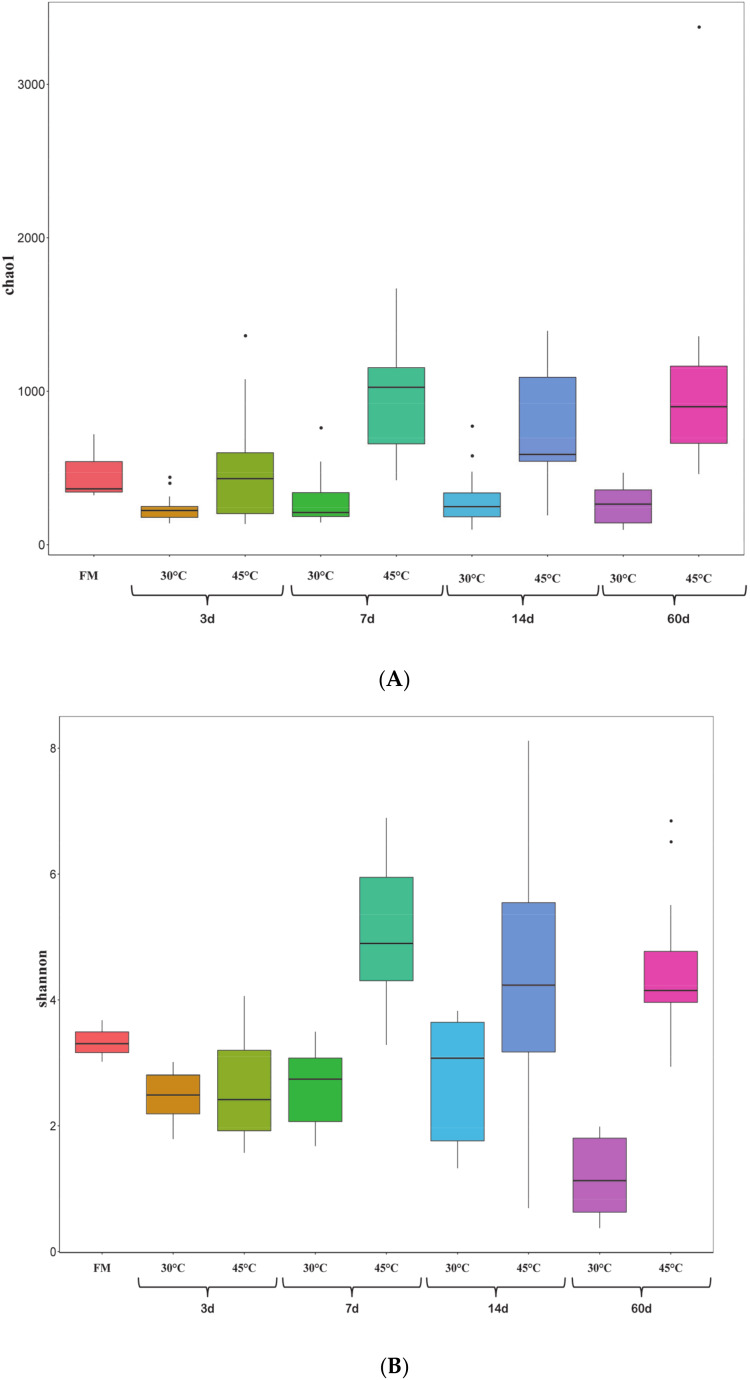
Community diversity and richness of the silages for every period: (**A**) Chao1 index of silage samples for every period (FM: fresh material; 3 d; 7 d; 14 d; 60 d); (**B**) Shannon index of silage samples for every period (FM: fresh material; 3 d; 7 d; 14 d; 60 d). Black dot means outlier.

**Table 1 microorganisms-08-00719-t001:** Fermentation characteristics of corn silage prepared with two temperatures at different ensiling period.

Item	30 °C					45 °C								
	Control	LS358	LR753	LS	LR	Control	LS358	LR753	LS	LR	SEM	Temperature	Inoculants	Interaction
3 d														
pH	3.91 ^d^	3.88 ^de^	3.87 ^e^	3.86 ^e^	3.81 ^f^	4.02 ^b^	4.02 ^bc^	3.98 ^c^	4.10 ^a^	4.00 ^bc^	0.018	**	**	**
NH_3_-N (g kg^−1^ TN)	23.82 ^ab^	15.08 ^de^	13.88 ^e^	17.94 ^cde^	17.89 ^cde^	25.04 ^a^	19.47 ^bcd^	15.15 ^de^	15.38 ^de^	20.64 ^bc^	1.458	NS	**	NS
LA (g kg^−1^ DM)	60.94 ^abc^	47.43 ^cd^	41.46 ^d^	74.12 ^a^	73.14 ^a^	50.86 ^bcd^	54.61 ^bcd^	60.93 ^abc^	65.07 ^ab^	64.88 ^ab^	6.357	**	NS	NS
AA (g kg^−1^ DM)	0	0	0	0	0	0	0	0	0	0	-	-	-	-
LA/AA	0	0	0	0	0	0	0	0	0	0	-	-	-	-
7 d														
pH	3.94 ^e^	3.94 ^e^	3.92 ^e^	3.93 ^e^	3.93 ^e^	4.19 ^a^	4.07 ^cd^	4.10 ^bc^	4.15 ^b^	4.02 ^d^	0.022	**	**	**
NH_3_-N (g kg^−1^ TN)	25.80 ^a^	25.57 ^a^	24.40 ^abc^	24.84 ^abc^	22.50 ^bcde^	25.38 ^ab^	22.05 ^cde^	20.50 ^e^	21.06 ^de^	20.24 ^e^	1.073	**	*	NS
LA (g kg^−1^ DM)	63.78 ^cde^	69.37 ^cde^	75.07 ^cd^	75.01 ^cd^	94.18 ^a^	88.35 ^ab^	78.83 ^bc^	63.13 ^e^	67.23 ^cde^	71.94 ^cde^	3.609	**	NS	**
AA (g kg^−1^ DM)	32.18 ^cd^	40.15 ^b^	55.87 ^a^	33.49 ^bcd^	37.93 ^bc^	0	30.26 ^d^	28.76 ^d^	0	0	2.289	**	**	**
LA/AA	1.99 ^b^	1.73 ^bc^	1.34 ^c^	1.82 ^bc^	1.77 ^bc^	0	2.76 ^a^	2.21 ^b^	0	0	0.168	**	**	**
14 d														
pH	3.92 ^c^	3.92 ^c^	3.90 ^c^	3.90 ^c^	3.86 ^c^	4.28 ^a^	3.91 ^c^	3.82 ^c^	4.04 ^b^	4.02 ^b^	0.028	**	**	**
NH_3_-N (g kg^−1^ TN)	32.01 ^ab^	29.60 ^bcd^	24.87 ^d^	27.53 ^bcd^	25.98 ^d^	36.02 ^a^	26.87 ^cd^	25.47 ^d^	31.41 ^bc^	27.95 ^bcd^	0.451	NS	**	NS
LA (g kg^−1^ DM)	61.16 ^d^	70.53 ^bcd^	70.82 ^bcd^	58.93 ^d^	102.06 ^a^	71.73 ^bcd^	86.43 ^abc^	87.03 ^ab^	85.57 ^abc^	65.07 ^cd^	6.478	NS	NS	**
AA (g kg^−1^ DM)	31.83 ^d^	47.52 ^b^	61.22 ^a^	38.97 ^c^	31.09 ^d^	25.85 ^d^	24.85 ^d^	28.43 ^d^	0	25.77 ^d^	2.379	**	**	**
LA/AA	1.95 ^bc^	1.48 ^c^	1.15 ^c^	1.51 ^c^	3.38 ^a^	3.38 ^a^	3.47 ^a^	2.54 ^ab^	0	2.63 ^ab^	0.299	*	**	**
60 d														
pH	3.84 ^de^	3.85 ^de^	3.85 ^de^	3.81 ^e^	3.87 ^cd^	4.06 ^a^	3.96 ^b^	3.91 ^c^	4.00 ^b^	3.91 ^c^	0.015	**	**	**
NH_3_-N (g kg^−1^ TN)	42.58 ^bc^	36.56 ^d^	25.81 ^e^	36.78 ^d^	30.03 ^e^	47.75 ^a^	46.88 ^ab^	37.77 ^cd^	40.62 ^cd^	39.29 ^cd^	1.632	**	**	**
LA (g kg^−1^ DM)	68.68 ^bcd^	66.72 ^cd^	63.01 ^de^	51.50 ^e^	72.23 ^abcd^	83.39 ^a^	84.63 ^a^	80.06 ^ab^	77.77 ^abc^	66.45 ^cd^	4.00	**	NS	**
AA (g kg^−1^ DM)	30.98 ^bc^	35.88 ^ab^	47.01 ^a^	21.73 ^cd^	35.56 ^ab^	18.43 ^d^	17.46 ^d^	33.40 ^bc^	21.28 ^cd^	31.17 ^bc^	3.81	**	**	*
LA/AA	2.21 ^bc^	1.88 ^bc^	1.34 ^bc^	2.43 ^bc^	2.73 ^b^	4.53 ^a^	4.85 ^a^	2.40 ^bc^	4.06 ^a^	1.93 ^bc^	0.403	**	**	**

^a–e^ Means with different superscripts within a row are significantly different (*p* < 0.05). * *p* < 0.05; ** *p* < 0.01; LS358 = *Lactobacillus salivarius* 358; LR753 = *Lactobacillus rhamnosus* 753; ATCC: American Type Culture Collection; LS = *Lactobacillus salivarius* ATCC 11741; LR = *Lactobacillus rhamnosus* ATCC 7469; LA = lactic acid; AA = acetic acid; LA/AA = ratio of lactic acid to acetic acid; NS = no significance; - = not detected.

**Table 2 microorganisms-08-00719-t002:** Chemical composition of corn silage ensiled for 60 d at 30 and 45 °C.

Item	30 °C					45 °C								
	Control	LS358	LR753	LS	LR	Control	LS358	LR753	LS	LR	SEM	Temperature	Inoculants	Interaction
DMR%	97.03 ^a^	96.24 ^a^	96.53 ^a^	96.61 ^a^	96.19 ^a^	82.22 ^b^	82.11 ^b^	84.13 ^b^	82.85 ^b^	84.51 ^b^	1.649	**	NS	NS
CP g kg^−1^ DM	69.16 ^c^	73.54 ^b^	64.37 ^e^	66.66 ^d^	75.62 ^a^	69.37 ^c^	67.08 ^d^	63.64 ^e^	56.70 ^f^	64.58 ^e^	0.922	**	**	**
WSC g kg^−1^ DM	15.47 ^e^	18.64 ^e^	14.85 ^e^	20.65 ^e^	55.89 ^d^	140.47 ^a^	119.40 ^bc^	108.48 ^c^	119.73 ^bc^	123.75 ^b^	5.767	**	**	**
NDF g kg^−1^ DM	446.14 ^ab^	450.77 ^ab^	509.23 ^a^	428.86 ^b^	484.98 ^ab^	443.22 ^ab^	434.32 ^b^	436.83 ^b^	454.70 ^ab^	437.04 ^b^	1.857	**	NS	*
ADF g kg^−1^ DM	201.29 ^ab^	208.39 ^abc^	231.09 ^a^	184.66 ^c^	220.71 ^ab^	211.35 ^abc^	198.67 ^bc^	183.05 ^c^	204.17 ^abc^	223.21 ^ab^	1.303	NS	NS	*

^a–e^ Means with different superscripts within a row are significantly different (*p* < 0.05). * *p* < 0.05; ** *p* < 0.01. LS358 = *Lactobacillus salivarius*358; LR753 = *Lactobacillus rhamnosus*753; LS = *Lactobacillus salivarius* ATCC 11741; LR = *Lactobacillus rhamnosus* ATCC 7469; DMR = dry matter recovery; CP = crude protein; WSC = water-soluble carbohydrates; NDF = neutral detergent fiber; ADF = acid detergent fiber; NS = no significant.

**Table 3 microorganisms-08-00719-t003:** Statistical analysis for the operational taxonomic units (OTUs) of corn silages treated with temperature and inoculants at different fermentation periods.

Period	Factor	Anosim		Adonis					
		R-Value	*p*-Value	Df	SumsOfSqs	MeanSqs	F.Model	R2	Pr(>F)
3 d	Temperature	0.5163	0.001	1(28)	2.2129(5.4017)	2.21285(0.19292)	11.47	0.29061(0.70939)	0.001
	Inoculants	0.3225	0.007	1(28)	1.1866(6.4279)	1.18665(0.22957)	5.1691	0.15584(0.84416)	0.004
7 d	Temperature	0.3763	0.002	1(28)	1.7292(7.2746)	1.72916(0.25981)	6.6556	0.19205(0.80795)	0.001
	Inoculants	−0.1115	0.871	1(28)	0.5382(8.4662)	0.53816(0.30236)	1.7799	0.05977(0.94023)	0.11
14 d	Temperature	0.4865	0.001	1(28)	2.3373(7.7235)	2.33728(0.27584)	8.4733	0.23231(0.76769)	0.001
	Inoculants	−0.1529	0.916	1(28)	0.2902(9.7711)	0.29017(0.34897)	0.8315	0.02884(0.97116)	0.575
60 d	Temperature	0.999	0.001	1(28)	5.3947(2.2199)	5.3947(0.0793)	68.045	0.70847(0.29153)	0.001
	Inoculants	−0.1173	0.974	1(28)	0.0829(7.5319)	0.082906(0.268995)	0.30821	0.01089(0.98911)	0.752

Anosim: Analysis of similarities; Adonis: Permutational multivariate analysis of variance, PERMANOVA (Bray-Curtis distance); Df: Degree of freedom; SumsOfSqs: Sums of squares; MeanSqs: Mean squares; R2: Variation (R2).

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
