# Peer review of "Microbial Community and Fermentation Dynamics of Corn Silage Prepared with Heat-Resistant Lactic Acid Bacteria in a Hot Environment"

_microorganisms, 2020, doi:10.3390/microorganisms8050719_

Round 1
Reviewer 1 Report
In the manuscript "Microbial community and fermentation dynamics of corn silage prepared with heat-resistant lactic acid bacteria in hot environment", Guan et al. describe how the use of two thermotolerant and two non-thermotolerant lactobacilli affected corn ensiling. While the obtained results are interesting, the manuscript would benefit from a clearer presentation.
Major remarks:
1) The manuscript does not state where did the thermotolerant lactobacilli come from. Where were they isolated, how do we know they are thermotolerant? The answers to these questions should be clearly stated in the Materials and Methods or the Results section. What is the rationale for using one homofermentative and one heterofermentative species? How does thermotolerant isolate of the species compare with the type strain? Why might that be so? These questions should be discussed in the separate paragraph of the Discussion.
2) In the Methods section "Bacterial community analysis", the authors state that microbe populations (presumably from sequencing experiment) were estimated as colony-forming units but provide no protocol. Did they mean instead to include subsection "Statistical analyses" under "Chemical, fermentation and microbial composition analysis"?
3) The Results section often just transcribes numbers readily available in the manuscript's tables, the extreme example being the subsection "Chemical and microbial composition of fresh forage" which just reiterates Table 1. Such repetition is unnecessary and unhelpful. The next subsection "Fermentation characteristics and microbial composition of silage at different period" then describes in minute details Tables 2 and 3. While it is important to present exact numbers, the manuscript would greatly benefit from several graphs clearly showing the observed trends, thus interpretable in mere seconds. If this were done correctly, some parts of Tables 2 and 3 could even be relegated to Supplementary materials. As it is, the reader needs to strenuously look at the tables and compare two numbers by two numbers, an unnecessarily long and tiring process. Moreover, in Tables 2 and 3, superscripts a-e are of the same size as numbers, making it unnecessarily cumbersome to discern where the number ends and letters begin.
4) In the present form, Figure 1 is unhelpful. It stretches outside the page, the data for the 60th day unavailable. The legend is missing as well, as do the explanations for abbreviations bellow the columns (e.g. CK). The figure's ratio is maladjusted.
5) The code used to process the NGS data and produce Figures 1 and 2 must be made readily available as Supplementary materials.
Minor comments:
5) There are several misspelt words (e.g. line 43: Duo = Due, line 95: quakily = quickly, line 119: Bole = Bile).
6) In the abstract, abbreviation DM is mentioned without being defined. It should be exchanged for the full term.
7) The last paragraph of the introduction would benefit from being expanded to present a more detailed list of what will be presented in the Results section.
8) The authors should provide a sequence and a reference for the 515F and 806R primers.
9) Some of the references are missing DOI, although these references have DOI assigned to them.
Author Response
- Reviwer1:
In the manuscript "Microbial community and fermentation dynamics of corn silage prepared with heat-resistant lactic acid bacteria in hot environment", Guan et al. describe how the use of two thermotolerant and two non-thermotolerant lactobacilli affected corn ensiling. While the obtained results are interesting, the manuscript would benefit from a clearer presentation.
Major remarks:
- The manuscript does not state where did the thermotolerant lactobacilli come from. Where were they isolated, how do we know they are thermotolerant? The answers to these questions should be clearly stated in the Materials and Methods or the Results section. What is the rationale for using one homofermentative and one heterofermentative species? How does thermotolerant isolate of the species compare with the type strain? Why might that be so? These questions should be discussed in the separate paragraph of the Discussion.
Response: Thank you so much for your advice. We have added more details about thermotolerant lactobacilli in Materia and Method and Discussion sections. “Two strains L. salivarius 358 (LS358) and L. rhamnosus 753 (LR753) were isolated from forage crops and silages in hot and humid area of southwest China (Sichuan Province, Chongqing city, Guizhou Province). Both strains can grow at a high temperature and a low pH conditions, and produce more lactic acid than other inoculant strains and isolates in silage environments. According to the screening results of physiological and biochemical properties and small-scale silage fermentation test of more than 200 LAB isolates, the LAB strains were selected for improving silage fermentation and inhibiting aerobic deterioration of silage (Guan et al., 2020a).” in lines 92-99.
- In the Methods section "Bacterial community analysis", the authors state that microbe populations (presumably from sequencing experiment) were estimated as colony-forming units but provide no protocol. Did they mean instead to include subsection "Statistical analyses" under "Chemical, fermentation and microbial composition analysis"?
Response: To distinguish the difference between the microbial composition and the microbial community in Materia and Method section, we changed the headline of ‘microbial composition’ to ‘microbial counts’ in line 116. The count data came from the traditional plate culture method, as mentioned in lines 133-136.
- The Results section often just transcribes numbers readily available in the manuscript's tables, the extreme example being the subsection "Chemical and microbial composition of fresh forage" which just reiterates Table 1. Such repetition is unnecessary and unhelpful. The next subsection "Fermentation characteristics and microbial composition of silage at different period" then describes in minute details Tables 2 and 3. While it is important to present exact numbers, the manuscript would greatly benefit from several graphs clearly showing the observed trends, thus interpretable in mere seconds. If this were done correctly, some parts of Tables 2 and 3 could even be relegated to Supplementary materials. As it is, the reader needs to strenuously look at the tables and compare two numbers by two numbers, an unnecessarily long and tiring process. Moreover, in Tables 2 and 3, superscripts a-e are of the same size as numbers, making it unnecessarily cumbersome to discern where the number ends and letters begin.
Response: Thank you so much for your advice. We deleted Table 1 and presented the chemical composition data of forage in Materials and Methods in lines 89-92. And we also changed data in Table 3 to four graphs (Figure1a,b,c,d) to reflect the dynamic changes in the number of lactic acid bacteria and yeasts under two temperature conditions. To better reflect the statistical data, we uploaded previous version of Table 3 as supplementary material T1. And we made a-e as superscript to distinguish with numbers.
- In the present form, Figure 1 is unhelpful. It stretches outside the page, the data for the 60th day unavailable. The legend is missing as well, as do the explanations for abbreviations bellow the columns (e.g. CK). The figure's ratio is maladjusted.
Response: Thank you so much for your advice. We have made adjustments and added the explanations for abbreviations bellow the columns.
- The code used to process the NGS data and produce Figures 1 and 2 must be made readily available as Supplementary materials.
Response: As mentioned in lines 162-163, the sequence data reported in this study have been submitted to NCBI database (PRJNA606702).
Minor comments:
- There are several misspelt words (e.g. line 43: Duo = Due, line 95: quakily = quickly, line 119: Bole = Bile).
Response: Thank you so much for your careful review, we have revised words.
- In the abstract, abbreviation DM is mentioned without being defined. It should be exchanged for the full term.
Response: Thank you so much for your careful review, we have exchanged to the full term.
- The last paragraph of the introduction would benefit from being expanded to present a more detailed list of what will be presented in the Results section.
Response: We have added more details in the last paragraph of the introduction as follow: “To develop silage fermentation technique in response to global warming, current study was to apply NGS to determine the dynamic (0, 3, 7, 14, 60 d of ensiling) changes of microbial community and fermentation profile of corn silage prepared with two screened heat-resistant LAB (Lactobacillus salivarius 358 and L. rhamnosus 753) in hot environment (30 °C and 45 °C) in lines 79-82.
- The authors should provide a sequence and a reference for the 515F and 806R primers.
Response: We have added the sequence and reference for the 515F and 806R primers in lines 148-149.
- Some of the references are missing DOI, although these references have DOI assigned to them.
Response: We have checked the DOI of references and added them.

Reviewer 2 Report
The manuscript of Guan et al. 2020, describes the effect of lactic acid bacteria strains and different temperatures of fermentation on chemical and microbiological features of corn silages. Changes in microbial composition and dynamism were also evaluated. The topic of manuscript is interesting for the reader of “Microorganisms”, but the paper needs several revisions before publication.
Introduction
- Please avoid the possessive form ‘s or abbreviations such as “can’t” in the manuscript
- more information about the use of LAB as inoculants during silage fermentation should added in the "Introduction sections". In the literature, many papers are available
Materials and Methods
- Please specify the features of selected LAB (Lactobacillus rhamnosus 753 and L. salivarius 358)? Why did the authors choose these strains? Were suitable for the fermentation of silage?
- The strains were inoculated as single culture or in combinations? Please specify, as this information is missing.
- How many biological replicates were carried out? The Authors used a control silage? Please specify.
- Why did the authors choose these sampling times (3, 7, 14, 60 days)? Would an additional time have been appropriate?
- Line 99: What mean “fresh samples”? The samples at time zero? How many samples? Control and inoculated silage?
- Please add the primer sequence. Results and Discussion
- Table 1 indicates the chemical and microbial features of un-inoculated silage? Please use the term “control” or “un-inoculated” to avoid confusion.
- Table 1: please use symbols in the notes of Table and not numbers to avoid confusion in the brackets
- Table 2: Analysis of variance is not clear. Please revise notes of Table 2 and make sure to perform an analysis that, for each parameter, allows to see better: 1) the effect of the strains within the same temperature, 2) the effect of the temperature for the same strain. Please write the letters of significance as superscript
- Table 3: please see comments to Table 2
- In my opinion, the results are a simple description of the obtained data obtained, but actually they should be reported in a more critical way; the data shown in Tables 2 and 3 are many, but they are difficult to correlate in this way; I suggest doing a PCA, or other correlation analyses and graphs to highlight the effect of inoculants and temperatures on the different parameters measured for each sample.-
- Consequently the results should also be rearranged
- Also, the effect of inoculants and T°C on the microbiota composition and dynamism should be addressed better.
- Figure 1: I can't see the complete figure from the pdf that I downloaded ...
- In this manuscript, the main focus is not clear, or is not properly described.
- The Authors want to evaluate the effect of LAB strains and T ° C of fermentation on the chemical and microbiological features (including the composition of the microbiota) of samples? If yes, both Results and Discussion should be revised and rearranged.
- Additionally, since in literature are present many papers on the effect of LAB strain (both homofermentative and heterofermentative), a better comparison should be provided in Discussion section.
- Please revise also "Conclusions" according to the overall changes on manuscript
Author Response
- Reviwer2:
The manuscript of Guan et al. 2020, describes the effect of lactic acid bacteria strains and different temperatures of fermentation on chemical and microbiological features of corn silages. Changes in microbial composition and dynamism were also evaluated. The topic of manuscript is interesting for the reader of “Microorganisms”, but the paper needs several revisions before publication.
Introduction
- Please avoid the possessive form ‘s or abbreviations such as “can’t” in the manuscript more information about the use of LAB as inoculants during silage fermentation should added in the "Introduction sections". In the literature, many papers are available
Response: Thank you so much for your advice. We have revised “can’t” to “cannot” in line 71. And we also added more information about use of LAB as inoculants as follow: “Generally, to ensure that silage enters the fermentation stage dominated by LAB as soon as possible, LAB inoculants are commonly used in silage production (Weinberg and Muck, 1996). Typical homofermentative LAB strain L. plantarum can inhibit the growth of microbe in silage by accelerating the decrease of pH (Keshri et al., 2018), while L. buchneri as most used heterofermentative LAB strain can produce acetic acid to inhibit the growth of yeast and mold, thus improving silage aerobic stability (Elferink et al., 1999). However, due to these two commercial LAB inoculants were screened from temperate regions (McDonald et al., 1991), which were mainly used for cold-season forages and corn to produce silage (McDonald et al., 1991, Whiter and Kung, 2001, Reich and Kung Jr, 2010), their effects in tropical and subtropical regions may be limited. Chen et al. (2013) reported that commercial LAB (L. plantarum and Pediococcus acidilactici) had no positive effect on Ryegrass silage at 45 °C. The effect of inoculating commercial LAB (L. plantarum MTD-1) was also limited in Napier grass silage which stored at 50 °C (Gulfam et al., 2017). Guan et al. (2020b) found that the commercial L. plantarum not only failed to improve the silage quality, but also accelerated the aerobic deterioration in corn silage stored at 35-40 °C. These studies indicate that commercial LAB which mostly screened from temperate regions have limited adaptability to high temperature environments.” in lines 55-69.
Materials and Methods
- Please specify the features of selected LAB (Lactobacillus rhamnosus 753 and L. salivarius 358)? Why did the authors choose these strains? Were suitable for the fermentation of silage?
Response: We added more details about these two stains as follow: “Two strains L. salivarius 358 (LS358) and L. rhamnosus 753 (LR753) were isolated from forage crops and silages in hot and humid area of southwest China (Sichuan Province, Chongqing city, Guizhou Province). Both strains can grow at a high temperature and a low pH conditions, and produce more lactic acid than other inoculant strains and isolates in silage environments. According to the screening results of physiological and biochemical properties and small-scale silage fermentation test of more than 200 LAB isolates, the LAB strains were selected for improving silage fermentation and inhibiting aerobic deterioration of silage (Guan et al., 2020a).” in lines 92-99.
- The strains were inoculated as single culture or in combinations? Please specify, as this information is missing.
Response: We have added more details as follow: “Four LAB strains were inoculated as single culture by using MRS broth (CM 188, Land Bridge Brands, Beijing, China)” in lines 101-102.
- How many biological replicates were carried out? The Authors used a control silage? Please specify.
Response: As mentioned in line 107, “Each treatment had three biological replicates”. And we used sterile distilled water as control silage in line 99.
- Why did the authors choose these sampling times (3, 7, 14, 60 days)? Would an additional time have been appropriate?
Response: Thank you so much for your advice. These time points well reflected the changes of microbial community and fermentation profile during ensiling.
- Line 99: What mean “fresh samples”? The samples at time zero? How many samples? Control and inoculated silage?
Response: Fresh sample means at time zero, we have revised as “Three fresh samples and all 60-day silage samples” in line 117.
- Please add the primer sequence.
Response: We have added the primer as follow: “16S rRNA genes of distinct regions (16S V4) were amplified used the specific primers 515F (5’-GTTTCGGT GCCAGCMGCCGCGGTAA-3’) and 806R (5’-GCCAA TGGACTACHVGGGTWTCTAAT-3’) (Guan et al., 2018)” in lines 148-149.
Results and Discussion
- Table 1 indicates the chemical and microbial features of un-inoculated silage? Please use the term “control” or “un-inoculated” to avoid confusion.
Response: Thank you so much for your advice. Combined with the comments of another reviewer, we deleted Table 1 and displayed the data of the chemical composition in Table 1 in the materials and methods in lines 89-92. We used term “Control” to represent the treatment adding equal amount of sterile distilled water.
- Table 1: please use symbols in the notes of Table and not numbers to avoid confusion in the brackets
Response: Thank you so much for your advice. We have revised them.
- Table 2: Analysis of variance is not clear. Please revise notes of Table 2 and make sure to perform an analysis that, for each parameter, allows to see better: 1) the effect of the strains within the same temperature, 2) the effect of the temperature for the same strain. Please write the letters of significance as superscript.
Response: Thank you so much for your advice. We made a two-way ANOVA at the end of the table as follow:
- Table 3: please see comments to Table 2
Response: Thank you so much for your advice. Combined with the comments of another reviewer, we changed data in Table 3 to four graphs (Figure 1 a,b,c,d) to reflect the dynamic changes in the number of lactic acid bacteria and yeasts under two temperature conditions. To better reflect the statistical data, we uploaded previous vison of Table 3 as supplementary material T1.
- In my opinion, the results are a simple description of the obtained data obtained, but actually they should be reported in a more critical way; the data shown in Tables 2 and 3 are many, but they are difficult to correlate in this way; I suggest doing a PCA, or other correlation analyses and graphs to highlight the effect of inoculants and temperatures on the different parameters measured for each sample.
Response: Thank you so much for your advice. As shown in new version of Table 1, we made a two-way ANOVA at the end of the table, which can clearly reflect the effect of inoculants and temperatures on the different parameters measured for each sample.
- Consequently, the results should also be rearranged
Response: Thank you so much for your advice. We added the effect of inoculants and temperatures on the different parameters measured for each sample during different period as follow:
3 d: “Temperature, inoculants and their interaction had extremely significant effect on pH, number of LAB and yeast (P < 0.01).” in lines 181-182.
7 d: “Temperature, LAB and their interaction had extremely significant effect on pH, AA contents, LA/AA, number of LAB and yeast (P < 0.01).” in lines 193-194.
14 d: “Temperature, inoculants and their interaction had extremely significant effect (P < 0.01) on pH, AA contents, number of LAB and yeast.” in lines 205-207.
60 d: “Temperature, inoculants and their interaction had extremely significant effect on pH, NH3-N (g kg-1 TN), and LA/AA (P < 0.01).” in lines 215-217.
- Also, the effect of inoculants and T°C on the microbiota composition and dynamism should be addressed better.
Response: Thank you so much for your advice. We made new Table 3 to show the effect of inoculants and T°C on the microbiota composition and dynamism, and added related contents in results as follow: “As shown in Table 3, the effect of inoculants and temperature on the microbiota community during ensiling and differences in Bray-Curtis distance were analyzed by permutational multivariate analysis of variance (Anosim and PERMANOVA). Temperature has been significantly (P < 0.05) affecting the bacterial community in corn silage from 3 d to 60 d, of which the greatest effect was observed at 60 d (R-value = 0.999). The significant (P < 0.05) effect (P < 0.05) of the inoculants on the bacterial community only lasted to d 7, and there was no significant effect (P > 0.05) at 14 and 60 d of ensiling. Moreover, the R-value < 0 of the inoculants was observed at 7, 14, and 60 d of ensiling.” in lines 249-256.
- Figure 1: I can't see the complete figure from the pdf that I downloaded
Response: Combined with the comments of another reviewer, we deleted Table 1 and displayed the data of the chemical composition in Table 1 in the materials and methods in lines 89-92.
- In this manuscript, the main focus is not clear, or is not properly described.
Response: The main focus of this manuscript has been described as “To develop silage fermentation technique in response to global warming, current study was to apply NGS to determine the dynamic (0, 3, 7, 14, 60 d of ensiling) changes of microbial community and fermentation profile of corn silage prepared with two screened heat-resistant LAB (L. salivarius 358 and L. rhamnosus 753) in hot environment (30 °C and 45 °C).” in lines 79-82.
- The Authors want to evaluate the effect of LAB strains and T ° C of fermentation on the chemical and microbiological features (including the composition of the microbiota) of samples? If yes, both Results and Discussion should be revised and rearranged.
Response: Yes, In the results section, we described the effects of temperature and inoculants on the fermentation profile and microbial community at different time points, as well as the effects of temperature and inoculants on chemical composition at 60 days. In the discussion section, since temperature was an important factor that mainly affects the fermentation profile and microbial community, we focused on discussing how temperature affected the fermentation profile and microbial community. Inoculants as a main factor that produced acetic acid, and we also discussed.
- Additionally, since in literature are present many papers on the effect of LAB strain (both homofermentative and heterofermentative), a better comparison should be provided in Discussion section.
Response: Thank you so much for your advice. We added more discussion about LAB strain as inoculants in lines 302-312, and other added discussion in lines 317-324 and 351-355.
- Please revise also "Conclusions" according to the overall changes on manuscript
Response: We revised conclusions as follow: “The bacterial community of fresh whole-crop corn was found to be dominated by Leuconostoc, Klebsiella and Lactococcus spp. Lactobacilli were the prevalent bacteria in all silages at 30 °C, followed by acetobacter, while the bacterial communities became more complex at 45 °C at 60 d of ensiling. High temperature affected the dynamic changes of microorganisms in corn silage and resulted in a shift from homofermentative to heterofermentative LAB community which led to poor fermentation with more DM loss and higher NH3-N content. The heat-resistant strain L. rhamnosus 753 show potential possibility to respond to global warming for improving silage fermentation in tropics and subtropics. The monitoring bacterial dynamics by NGS technology is expected to better understand the relationships between microbial community and silage fermentation.” in lines 374-382.

Round 2
Reviewer 2 Report
The Authors addressed all questions. I endorse the